# Back to the Future: Revisiting the Application of an Enzyme Kinetic Equation to Maize Development Nearly Four Decades Later

**James R. Kiniry** [1,*] , **Sumin Kim** [2] **and Henri E. Z. Tonnang** [3]

1   USDA-ARS, 808 East Blackland Rd., Temple, TX 76502, USA
2   Oak Ridge Institute for Science and Education, 808 East Blackland Rd., Temple, TX 76502, USA; sumin.kim@usda.gov
3   International Institute of Tropical Agriculture (IITA), 08 BP 0932 Abomey Calavi, Cotonou, Benin; htonnang@gmail.com
*   Correspondence: jim.kiniry@ars.usda.gov; Tel.: +1-254-770-6506

**Abstract:** With the recent resurgence in interest in models describing maize (*Zea mays* L.) development rate responses to temperature, this study uses published data to refit the Poikilotherm equation and compare it to broken stick "heat stress" equations. These data were for the development rate of eight open pollinated maize varieties at diverse sites in Africa. The Poikilotherm equation was applied with the original published parameters and after refitting with the data in this study. The heat stress equation was tested after fitting with just the first variety and after fitting with each variety. The Poikilotherm equation with the original parameter values had large errors in predicting development rates in much of the temperature range. The adjusted Poikilotherm equation did much better with the root-mean-square error (RMSE) decreasing from 0.034 to 0.003 (1/day) for a representative variety. The heat stress equation fit to the first variety did better than the Poikilotherm equation when applied to all the varieties. The heat stress equations fitted separately for each variety did not have an improved fit compared to the one heat stress equation. Thus, separate fitting of such an equation for different varieties may not be necessary. The one heat stress equation, the separate heat stress equation, and the Poikilotherm equation each had a better fit than nonlinear Briere et al. curves. The Poikilotherm equation showed promise, realistically capturing the high, low, and optimum rate values measured. All the equations showed promise to some degree for future applications in simulating the maize development rate. When fitting separate regressions for each variety for the heat stress equations, the base temperatures had a mean of 5.3 °C, similar to a previously published value of 6 °C. The last variety had noticeably different rates than the others. This study demonstrated that a simple approach (the heat stress equation) should be adequate in many cases. It also demonstrated that more detailed equations can be useful when a more mechanistic system is desired. Future research could investigate the reasons for the different development rate response of the last variety and investigate similar varieties.

**Keywords:** phenology; crop modeling; degree days; heat units

## 1. Introduction

There has been a resurgence in interest in systems describing maize (*Zea mays* L.) development rate responses to temperature, as evidenced by recent studies by Kumudini et al. [1] and Tonnang et al. [2]. Both publications described several linear and nonlinear equations for maize development as a function of temperature.

The response of plant development rate to temperature has been studied and quantified for hundreds of years. In 1735, Reamur [3] developed the "thermal constant" idea. He summed daily temperature ranges and assumed the sum was stable between development stages. In the last 70 years, various empirical techniques have been used to calculate the growing degree days (GDD) [4–8]. These equations all assume a linear response of development rate above a base temperature.

Nonlinear equations for the development rate are not as common. Gilmore and Roger [9] developed a nonlinear equation for maize development rate using Lehenbauer's growth data [10]. Sharpe and DeMichael [11] made a giant step forward by attempting to develop a mechanistic, nonlinear equation for the development rate. In addition, Brier et al. [12] introduced a different form of nonlinear equation for the development rate.

Sharpe and DeMichele [11] combined reaction theories into a model to describe the three temperature response regions of development rate. This "Poikilotherm" equation contains six fitted variables that define how plant development responds to temperature. Kiniry and Keener [13] applied the equation with maize coleoptile extension rate data by Lehenbauer [10] to field data for maize development stages. The equation is based on the assumption of one control enzyme, reversibly denatured at high and low temperatures, being responsible for the nonlinear response. They assumed the control enzyme exists in these three states and the active enzyme was the intermediate. They developed the Poikilotherm (PK) equation, calculating the relative development rate ($R_t$) as a function of absolute temperature ($T$):

$$r(T) = \left[ \frac{T \cdot \exp\left[(\Delta S_A - \Delta H_A/T)/R\right]}{1 + \exp[(\Delta S_L - \Delta H_L/T)/R] + [(\Delta S_H - \Delta H_H/T)/R]} \right] \qquad (1)$$

where $T$ = absolute temperature (°K), $\Delta S_A$ = entropy of activation of the reaction, (19.49 cal/mol/°K), $\Delta H_A$ = enthalpy of activation of the reaction, (9170 cal/mol), $R$ = real gas constant, (1.9858775 cal/deg/mol), $\Delta S_L$ = entropy for the low-temperature denaturation, −95.70, $\Delta H_L$ = enthalpy for the low-temperature denaturation, (27,794 cal/mol), $\Delta S_H$ = entropy for the high-temperature denaturation, (211.8 cal/mol/°K), $\Delta H_H$ = enthalpy for the high-temperature denaturation, (65,203 cal/mol).

Rate ($R_t$) is a unitless value that varies from 0.0 at zero-development to 1.0 at the optimum temperature. This was the first time both high- and low-temperature denaturation were taken into account in one equation. These values for the parameters were the result of Sharpe and DeMichele fitting the equation to Lehenbauer's [10] constant temperature maize shoot elongation data. Tonnang et al. [2] included this enzyme kinetic equation in their comparisons with maize development in Africa.

The Briere model [12] is a nonlinear alternative to the Poikilotherm model. The model has four aspects: (1) The estimation of lower and upper temperature thresholds; (2) an asymmetric response around the optimum temperature; (3) an inflection point; and (4) a decay in development rate at supra-optimum temperatures. Tonnang et al. [2] were the first to apply the model to maize development rates (1/day). The lower and the upper thermal limits were integrated in the equation to represent parameters $T_b$ and $T_m$. To obtain the decay at high temperatures, a square root was included to allow a high slope when the temperature values approached $T_m$. By combining the products of different powers of temperature, an inflection point occurs, yielding the Briere_1 model as

$$r(T) = a \times T(T - T_o)\left( \sqrt{T_{max} - T} \right). \qquad (2)$$

A second model, Briere_2, was derived from the Briere_1 model by adding a general power equal to $d = 1/\mu$ to the square root term, where $\mu$ is a new parameter and $a$ is an empirical constant. The Briere_2 equation is

$$r(T) = a \times T(T - T_o)\left( \sqrt{T_{max} - T} \right)^d \qquad (3)$$

Parameters for each variety for these equations are given in Table 1. The Briere equations used were Briere_2 for the last variety (VE 220) and Briere_1 for all the others.

**Table 1.** Estimated parameters of Briere—1 and Briere—2 models for eight maize varieties (Tonnang et al. [2]).

| Vriety | | *Tb* (°C) | *Tm* (°C) | *a* |
|---|---|---|---|---|
| VE 201 | | 8.98 | 40.11 | 0.0003 |
| VE 203 | | 10.14 | 10.14 | 0.0003 |
| VE 206 | | 9.51 | 38.49 | 0.0003 |
| VE 208 | | 8.57 | 38.587 | 0.0003 |
| VE 210 | | 9.70 | 39.12 | 0.0003 |
| VE 212 | | 9.03 | 37.24 | 0.0003 |
| VE 218 | | 8.11 | 40.06 | 0.0029 |
| **Variety** | **μ** | ***Tb* (°C)** | ***Tm* (°C)** | ***a*** |
| VE 220 | 1.03 | 10.03 | 40.43 | 0.00004 |

The "heat stress" equation [9] assumes a linear response from the base temperature up to the optimum temperature. The development rate is assumed to decrease linearly above the optimum. In the present study, this "heat stress" or "broken stick" system is further refined. Additional work [14,15] resulted in deriving a base temperature of 8 °C for the maize development rate and an optimum temperature and decrease in the rate as temperatures exceeded the optimum. This "heat stress" equation was applied by Kiniry and Keener [13]. This work was then incorporated into the CERES-Maize model [16]. A lower base temperature of 6 °C was reported by Bonhomme et al. [17]. This heat stress equation is simply the temperature minus the base for temperatures above the base and below the optimum.

$$\text{Heat units} = T_{mean} - T_{base} \text{ if } T_{mean} \text{ is not above } T_{optimum}, \tag{4}$$

and

$$\text{Heat units} = (T_{optimum} - T_{base}) \times (1.0 - (T_{mean} - T_{optimum})/(T_{high} - T_{optimum}) \text{ if } T_{mean} \text{ is above } T_{optimum}, \tag{5}$$

where $T_{mean}$ is the mean daily temperature, $T_{base}$ is the base temperature (8 °C), $T_{optimum}$ is the optimum temperature, and $T_{high}$ is the high temperature where the development is zero, with all temperatures in °C.

Thus, as temperatures exceed the optimum, the rate decreases linearly. High temperatures above the optimum show decreases in the rate of development. When Kiniry and Keener [13] used the Poikilotherm equation with the growth data of Lehenbauer [10], it proved to be less consistent than the heat stress equation in simulating maize development rates in the field. Similarly, the recent Tonnang et al. [2] work showed that the nonlinear equations of Briere et al. [12] were the best of all 82 tested equations in the paper.

The objective of the present study was to take one more step in using the data of Tonnang et al. [2], attempting to refit the Poikilotherm equation originally fit with Lehenbauer's data, using data of the first maize variety (VE201) in the Tonnang et al. [2] study. We investigated how well this new fitted equation performed in describing the development of all the other varieties they measured. We also compared results with a fitted heat stress equation. The overall objective was to investigate the accuracy of both detailed, mechanistic equations with a simpler, easier-to-derive equation, all for predicting maize development rates. These results will give modelers guidance as to which approach is most suitable for meeting their goals. It will also provide estimates for the overall value of these more detailed models compared to the simpler heat stress equation.

## 2. Materials and Methods

The maize field data are for the development rate from seedling emergence to silking and the development rate from silking to physiological maturity [2]. It was assumed that the

temperature-dependent development rate of seedling emergence to silking and silking to physiological maturity was the same in any given variety based on previous experience of Tonnang et al. [2]. This is supported by the results of Kiniry and Keener [13]. In that study, assuming 4 days from planting to seedling emergence under optimum temperatures, the heat units from silking to physiological maturity for three hybrids were 53, 49, and 49% of the heat units from seedling emergence to physiological maturity. Thus, the temperature-dependent development rates from seedling emergence to silking were similar to the temperature-dependent development rates from silking to physiological maturity.

The data of Tonnang et al. [2] were for 9 medium-maturity open pollinated varieties that were described in Makumbi et al. [18]. These varieties are herbicide-resistant and show some resistance to *Striga hermonthica* (Del.) Benth. and *Striga asiatica* (L.) Kuntze that can severely affect some maize in sub-Saharan Africa. All plots were irrigated as needed to prevent drought stress.

The Poikilotherm equation, fitted with Lehenbauer's maize growth data, was adjusted using the relative development rates of Tonnang's first variety, VE 201. As discussed below and shown in Figure 1, the original parameters from Lehenbauer's growth data produced a curve that showed consistent bias at low and high temperatures. We addressed this in two steps: First, we fit the height of the curve to match the maximum development rate measured. Secondly, we adjusted the lateral scale (temperature) by visually fitting a factor affecting temperature. The temperature for the curve fitting was the measured temperature minus 8 °C, as described above. Thus, the temperature above the 8 °C base was divided by this factor. The use of 8 °C as a base was simply to be consistent with earlier work. This shifted the curve to the left, without changing the actual shape of the curve. It is worth noting that the original Sharpe and DeMichele publication contained an error in the final equation (Equation (17) in their publication). The second $\Delta H$ in the denominator should have had "H" as the subscript to denote that it was for high temperature rather than the "L" for low temperature.

Similarly, for the "heat stress" equation, the base temperature was assumed to be 8 °C. The development rate was linear from 8 °C to the optimum, which was the highest data point measured for variety VE 201. The development rate then decreased linearly from this optimum point through the one high temperature data point reported. In addition, the "heat stress" equation was fitted to their first variety data, using the established base temperature of 8 °C and the maximum measured development rate as the optimum rate. The linear decrease for above-optimum values was fitted to this maximum and the one datapoint at high temperatures. Finally, individual regression lines were fitted for each variety for all the data except the highest temperature data point. The base temperatures were derived for each and the regressions were used as another method of calculating development rate. For temperatures above the optimum, the line forced through the two highest temperature data points was used.

Both equations, derived for the first variety, were then applied without refitting to the data for the rest of the maize varieties presented in Tonnang et al. [2]. The root-mean-square error (RMSE) was used as a measure of the goodness of fit of each model for each maize variety [19,20].

$$RMSE = \sqrt{\frac{\sum_{i=1}^{n}\left(Y_i - \hat{Y}_i\right)^2}{n - K - 1}}, \tag{6}$$

where $n$ is the number of data points, $K$ is the number of model parameters, and $Y_i$ and $\hat{Y}_i$ are the observed and predicted values, respectively.

The best-fitted model is often selected by comparing Akaike's Information Criterion (AIC). However, as the sample size is small ($n/K < 40$), the small-sample (second-order) bias correction term, denoted by AICc, was calculated for each model within each maize variety [21,22].

$$AIC = n \times \ln\left(\frac{\sum_{i=1}^{n}\left(Y_i - \hat{Y}_i\right)^2}{n}\right) + 2K, \tag{7}$$

and

$$AIC_c = AIC + \frac{2K \times (K+1)}{n - K - 1}. \tag{8}$$

AICc differences, $\Delta_j\_AICc$, were also used to determine the level of support for each model [22]. The AICc with the minimum value, denoted by AICc_min, refers to the best model. The $\Delta_j\_AICc$ for the jth model is the difference between the $AICc_j$ and AICc_min, shown as follows:

$$\Delta_{j\_}AICc = AICc_j - AICc\_min. \tag{9}$$

There are some rough rules of thumb that are particularly useful for the nested model [22]. If the $\Delta_j\_AICc$ is

| $\Delta_j\_AICc$ | Level of support of Model j |
| --- | --- |
| 0–2 | There is substantial evidence that supports the model j. |
| 4–7 | The Model j has considerably less support. |
| >10 | The Model j has essentially no support. |

The Akaike weights, denoted by $w_j$, were used as a measure of the strength of evidence for each model; the Akaike weights count the probability that model j is the best among the set of models [22]. They are the ratio of each model's AICc differences relative to the sum of the AICc differences for all models, shown as follows:

$$w_j = \frac{e^{-0.5 \times \Delta_{j\_}AICc}}{\sum_{j=1}^{n} e^{-0.5 \times \Delta_{j\_}AICc}} \tag{10}$$

The values of RMSE, AICc, $\Delta_j\_AICc$, and $w_j$ for the Briere model were obtained and derived from the values reported in the original paper [2].

## 3. Results and Discussion

Development rates for all these analyses were for the development rate to a given growth stage, as described above. These were from field data and used the daily mean temperatures. The Poikilotherm equation with original Lehenbauer parameter values was not adequate (Figure 1). It had a much greater RMSE value (0.034) than the other equations for a representative variety (Figure 1 and Table 2). While not shown, results showed similar trends with all varieties. The Poikilotherm equation with original Lehenbauer parameter values underestimated the development rates below 25 °C and overestimated the rate at the high temperature. This lack of agreement with the development rate data is not unexpected. The Lehenbauer data were on growth rates and this is an application to development rates.

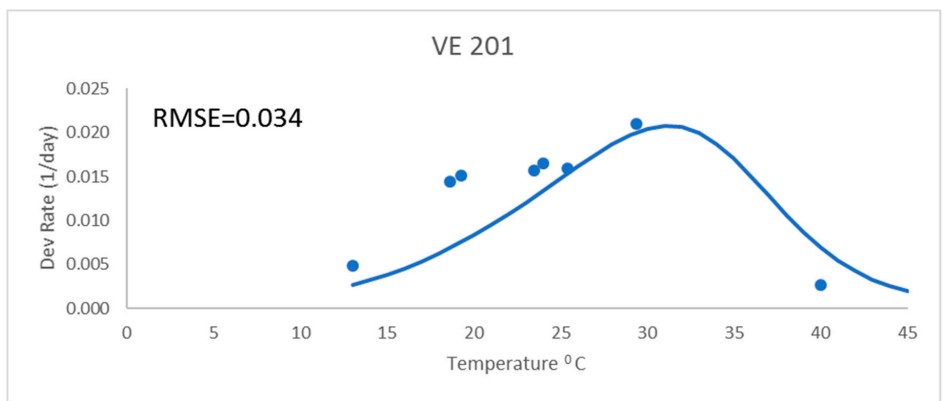

**Figure 1.** Poikilotherm equation with original parameters from Lehenbauer's maize coleoptile elongation rate data, applied to the first variety (VE 201). RMSE indicates the root-mean-square error for the maize variety's development rate.

**Table 2.** Results with root-mean-square error (RMSE) for the published equations of Tonnang et al. [2] (Briere), with the Poikilotherm (Pk) equation [11] fit as described herein, for the heat stress equation fitted for just the first variety (Htstr), and for the heat stress fitted separately for each variety (Indiv fitted); also shown are Akaike's Information Criterion (AIC), AICc, AICc differences ($\Delta_j$AICc), and the derived Akaike weights ($\omega_j$) computed from the data. The Tbase is the x intercept for each equation that was fitted for each variety, as discussed herein. The last variety (VE 220) was not included in these analyses as discussed in this paper. K indicates the number of parameters that were used to calculate the maize development rate. The AICc-selected best model is shown here in bold.

| Variety | Model | K | RMSE | AIC | AICc | $\Delta_j$AICc | $\omega_j$ |
|---------|-------|---|------|-----|------|----------------|-----------|
| VE 201 | **Htstr** | **2** | **0.003** | **−94.779** | **−92.379** | **0.000** | **0.720** |
| | Indiv fitted | 3 | 0.003 | −93.671 | −87.671 | 4.708 | 0.068 |
| | PK | 2 | 0.003 | −92.327 | −89.927 | 2.451 | 0.211 |
| | Brierea [a] | 3 | 0.001 | −74.339 | −68.339 | 24.039 | 0.000 |
| VE 203 | **Htstr** | **2** | **0.003** | **−91.961** | **−89.561** | **0.000** | **0.738** |
| | Indiv fitted | 3 | 0.003 | −91.320 | −85.320 | 4.241 | 0.088 |
| | PK | 2 | 0.004 | −89.073 | −86.673 | 2.888 | 0.174 |
| | Brierea [a] | 3 | 0.002 | −68.789 | −62.789 | 26.772 | 0.000 |
| VE 206 | **Htstr** | **2** | **0.003** | **−93.860** | **−91.460** | **0.000** | **0.885** |
| | Indiv fitted | 3 | 0.003 | −90.808 | −84.808 | 6.652 | 0.032 |
| | PK | 2 | 0.004 | −89.120 | −86.720 | 4.740 | 0.083 |
| | Brierea [a] | 3 | 0.002 | −71.788 | −65.788 | 25.672 | 0.000 |
| VE 208 | **Htstr** | **2** | **0.003** | **−93.433** | **−91.033** | **0.000** | **0.773** |
| | Indiv fitted | 3 | 0.003 | −91.182 | −85.182 | 5.851 | 0.041 |
| | PK | 2 | 0.000 | −90.579 | −88.179 | 2.854 | 0.186 |
| | Brierea [a] | 3 | 0.002 | −70.744 | −64.744 | 26.289 | 0.000 |
| VE 210 | **Htstr** | **2** | **0.003** | **−92.947** | **−89.947** | **0.000** | **0.819** |
| | Indiv fitted | 3 | 0.003 | −91.572 | −83.572 | 6.375 | 0.034 |
| | PK | 2 | 0.004 | −89.507 | −86.507 | 3.439 | 0.147 |
| | Brierea [a] | 3 | 0.002 | −70.381 | −64.381 | 25.566 | 0.000 |
| VE 212 | **Htstr** | **2** | **0.003** | **−94.621** | **−92.221** | **0.000** | **0.938** |
| | Indiv fitted | 3 | 0.003 | −91.415 | −85.415 | 6.805 | 0.031 |
| | PK | 2 | 0.004 | −87.810 | −85.410 | 6.810 | 0.031 |
| | Brierea [a] | 3 | 0.002 | −72.364 | −66.364 | 25.856 | 0.000 |
| VE 218 | **Htstr** | **2** | **0.003** | **−93.067** | **−90.667** | **0.000** | **0.689** |
| | Indiv fitted | 3 | 0.004 | −89.953 | −83.953 | 6.714 | 0.024 |
| | PK | 2 | 0.003 | −91.313 | −88.913 | 1.754 | 0.287 |
| | Brierea [a] | 3 | 0.002 | −70.996 | −64.996 | 25.671 | 0.000 |

[a] K, RMSE, and AIC values for Briere equation were obtained from Tonnang et al. [2], and values of AICc, $\Delta_j$AICc, and $\omega_j$ were computed based on the published values.

After adjusting the horizontal scale of this curve with a 1.04 factor (from visual fitting), the Poikilotherm equation did much better (Figure 2 and Table 2). Thus, the horizontal scale (temperature) was divided by this factor in the original equation to shift the predicted y values to the left. It was especially realistic at capturing the highest, lowest, and optimum rate values measured. This illustrates that the form of the equation could be useful for plant development, once fit to actual development rate data. Thus, the equation still holds some promise for application to the development of plants such as maize. The one obvious exception is the last variety, VE 220. Development rates below the optimum were greatly underpredicted by this general Poikilotherm equation.

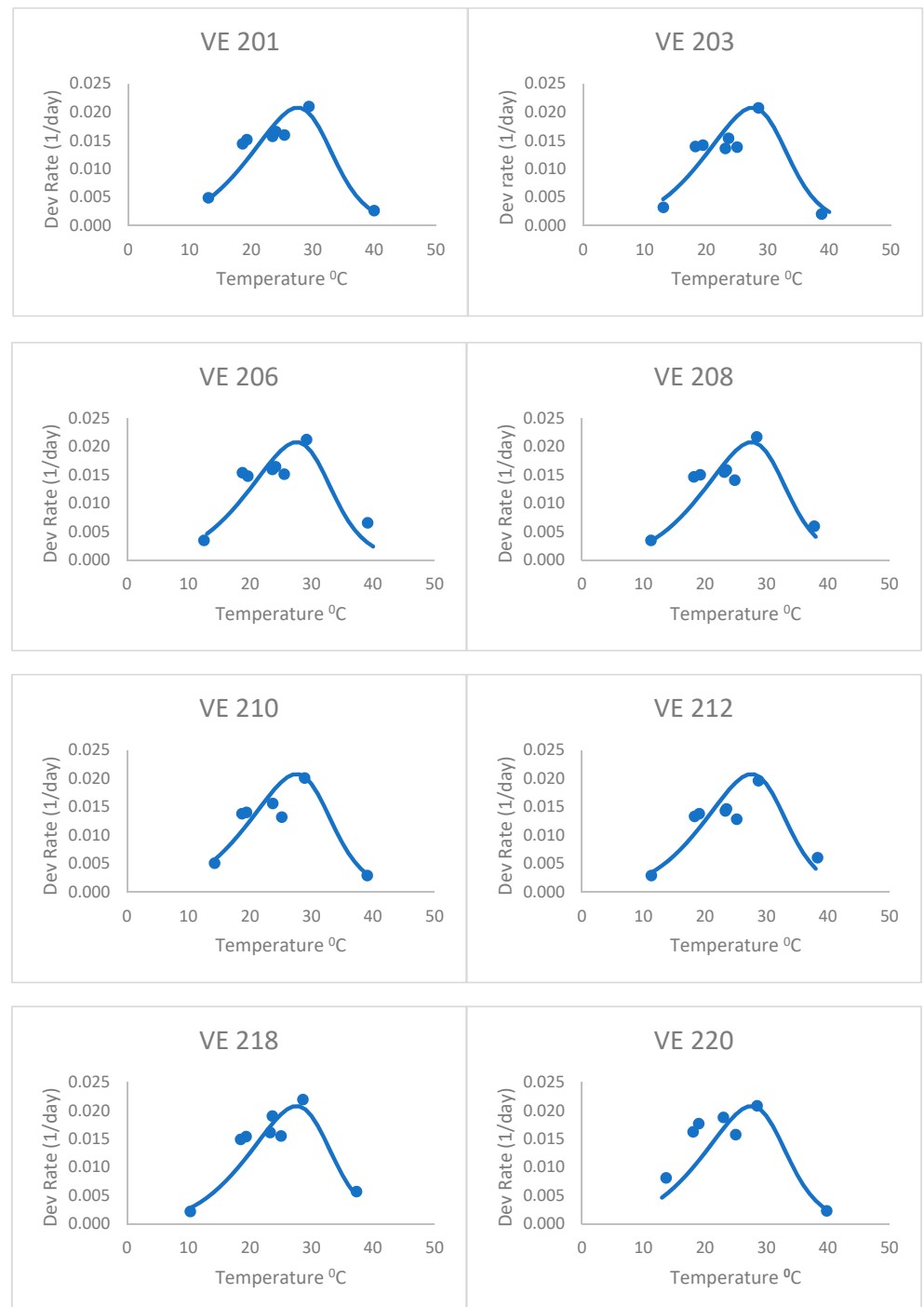

**Figure 2.** Poikilotherm equation fit to the first variety (VE 201) and applied to all varieties.

The heat stress equation (Figure 3), with an 8 °C base temperature and forced through the optimum and highest data point for the first variety, did better than the Poikilotherm equation. The heat stress equation had lower values of RMSE and AICc than the Poikilotherm equation (Table 2). This was fitted only using the data for the first variety, but still did a remarkable job on the other varieties as well. Based on the AICc value (Table 2), the heat stress equation was indicated as the best by all approaches. The AICc values for the heat stress equations ranged from −94.38 to −91.96 for all seven maize varieties (Table 2). In addition, the corresponding mean Akaike weight averaged over all maize varieties was 0.80, which means that given data, this equation has an 80% probability of being the best one. This success is similar to the results of Kiniry and Keener [13]. It is exciting to see that fitting the equation

with only one variety (VE 201) resulted in an equation that was reasonable for all except perhaps the last variety (VE 220). The RMSE for VE 220 (0.005, data not shown) was higher than the values of RMSE for other varieties (0.003) (Table 2).

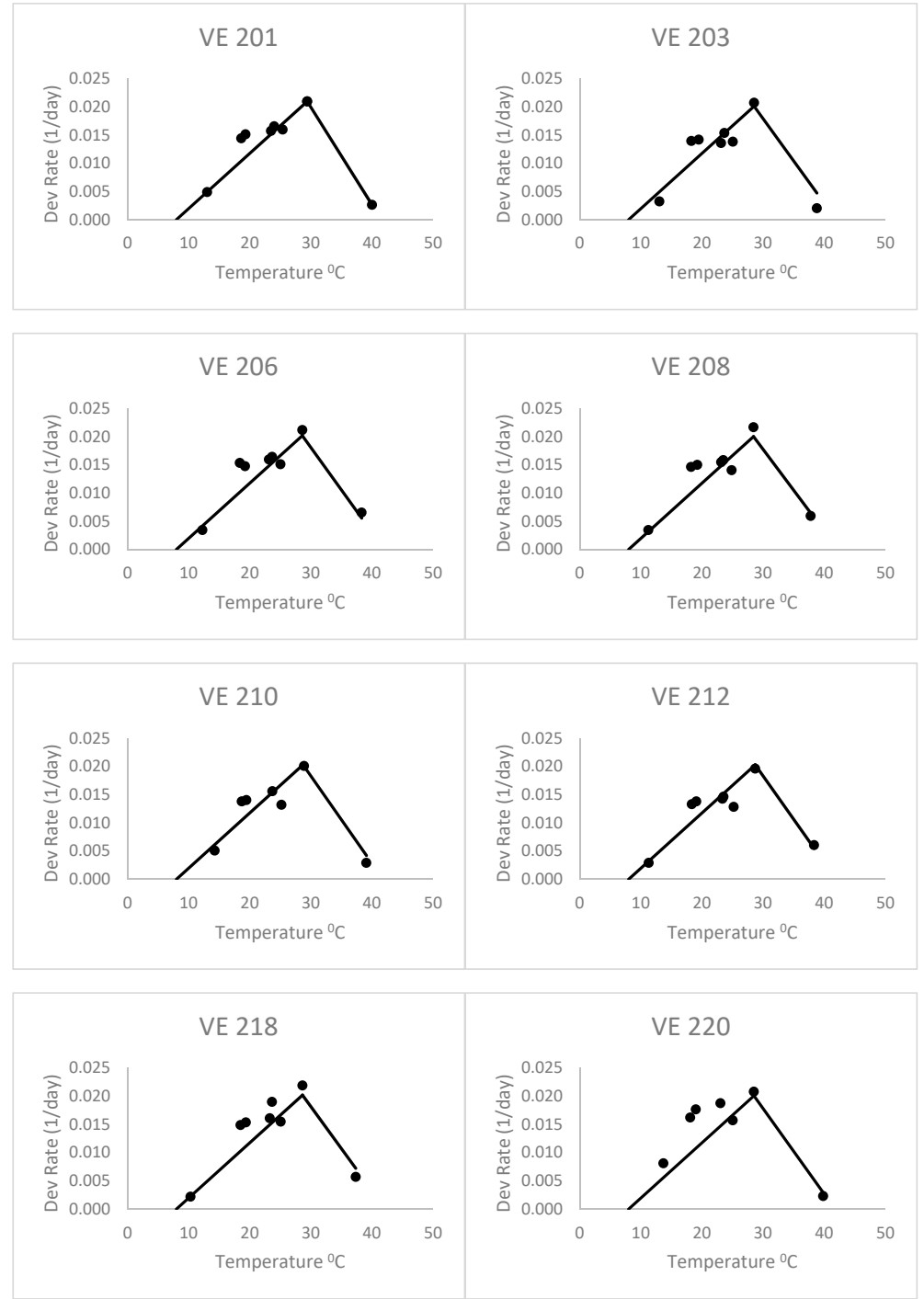

**Figure 3.** Heat stress regressions (forced through a base temp. of 8 °C) fit to the first variety (VE 201) and then applied to all the varieties.

When fitting separate regressions for each variety for the heat stress equations (Figure 4), the base temperatures (Tbase) ranged from 4.2 to 6.7 °C with a mean of 5.3 °C. This was without the last variety, which had a very low base temperature (lower than zero). Thus, the base temperatures of this group of varieties were lower than the commonly accepted base of 10 °C, but even lower than the

8 °C of Kiniry [14] and Kiniry and Bonhomme [15], discussed above. This value of 5.3 °C is similar to the Bonhomme et al. [17] value of 6 °C. As a result, the heat stress equations fitted separately for each variety (Figure 4) had better fits based on the RMSE and AICc (Table 2) than the heat stress equation. The mean $\Delta_j$_AICc over all maize varieties was 0.05, which was lower than the heat stress and Poikilotherm equations. This indicates that the heat stress equations fitted separately for each variety were not superior to the heat stress equation with the same set of parameters for all varieties. Thus, separate fitting of such an equation may not be necessary for a large number of varieties.

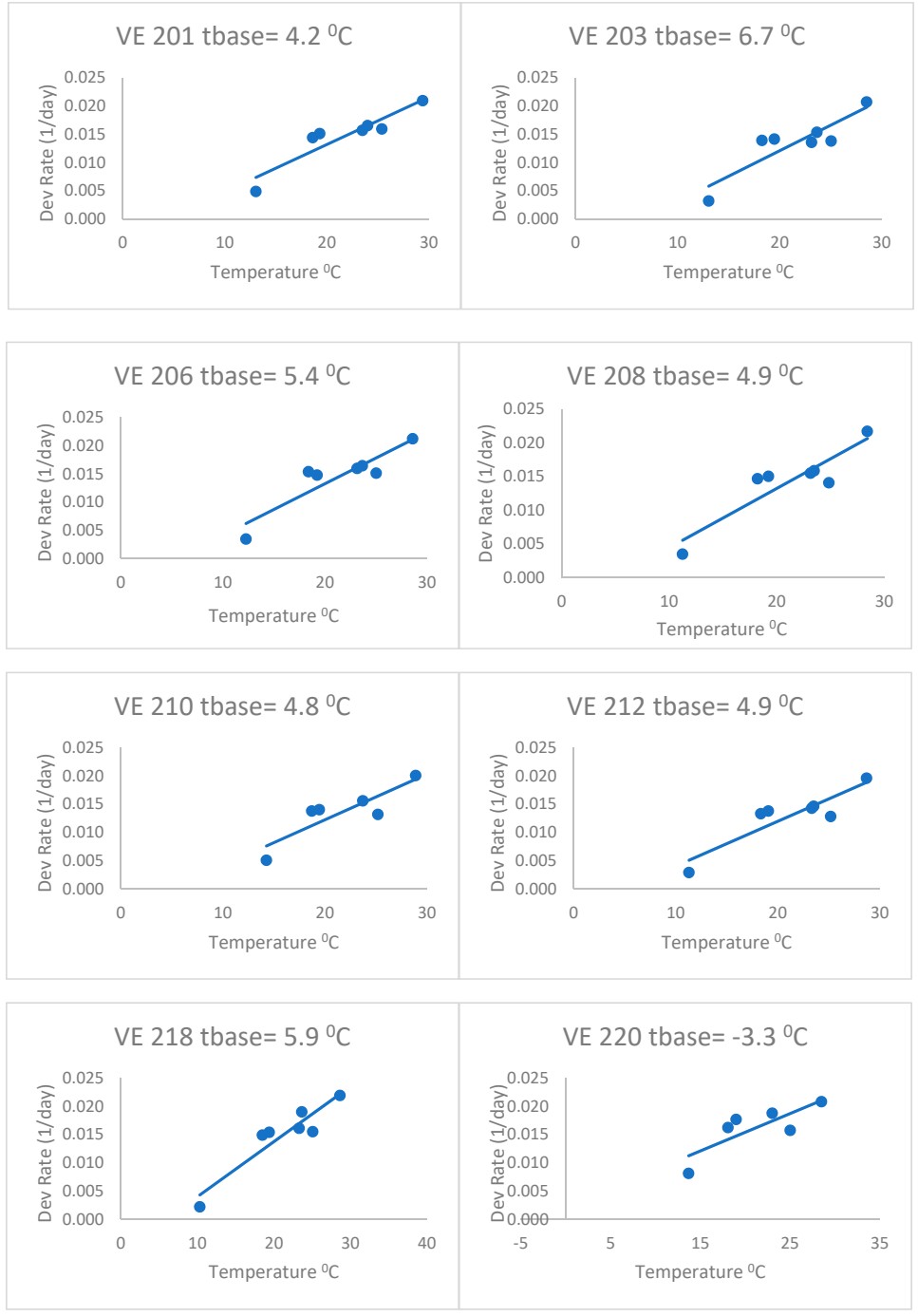

**Figure 4.** Regression fit to the data for each variety separately. These results are for the equation without the highest temperature data point.

As described above, the Briere equations used were Briere_2 for the last variety (220) and Briere_1 for all others. The Briere equations were fit to the data of each variety separately by Tonnang et al. [2]. Their parameters for the various varieties (excluding the last variety) showed base temperatures ranging from 8.1 °C to 10.1 °C with an average of 9.2 °C. The temperatures for the maximum development rates ranged from 37.2 °C to 40.1 °C with an average of 39.0 °C. According to Tonnang et al. [2], these were the best equations for each variety. However, in this study, based on the AICc value, the heat stress equation had a better fit than the Briere equations [2]. The mean $\Delta_j$_AICc over all maize varieties was 26, which means that the Briere_1 equation was unlikely to be the best model. It is apparent that the heat stress equation was closer to the measured values for the highest temperature data point, for the lower temperature data point, and for the data point with the greatest development rate.

This study demonstrated that a simple approach (the heat stress equation with one set of parameters) should be adequate in many cases. It also demonstrated that more detailed equations can be useful when a more mechanistic system is desired. The last variety (VE 220) had noticeably different rates than the others. Future research could investigate the reasons for this different development rate and investigate similar varieties to this. The difference from the other varieties is especially evident in the last heat stress analyses.

Studies of this type, using daily values for temperature within equations for development rate, always leave questions as to the source of errors in prediction. Plants are obviously responding to shorter time-frame temperatures such as hourly and often early in development when the growth point is below the soil surface, responding to soil temperatures instead of the measured air temperatures. Likewise, fitting development data with the means of fluctuating ambient temperatures potentially suffers from inaccuracies due to nonlinear responsiveness as described herein. This has led to several attempts at using controlled temperature growth chambers instead of field environments, as reviewed by Kiniry [16] and Kiniry and Bonhomme [15]. Even those studies must be interpreted with caution due to the possibility of unrepresentative conditions of light and air movement in such chambers. Future work on modeling maize phenology holds promise for exciting improvements, as many of these issues are addressed.

## 4. Conclusions

The heat stress equation with one set of parameters for all varieties was the best model. The heat stress equation fit to the first variety did better than the Poikilotherm equation when applied to all the varieties. The heat stress equations fitted separately for each variety did not improve the fit of each compared to the one heat stress equation and was inferior due to the greater number of fitted parameters. Thus, separate fitting of such an equation for different varieties may not be necessary. The one heat stress equation, the separate heat stress equation, and the Poikilotherm equation each had a better fit than the nonlinear Briere et al. curves. The Poikilotherm equation showed promise, realistically capturing the high, low, and optimum rate values measured. All the equations showed promise to some degree for future applications in simulating the maize development rate. When fitting separate regressions for each variety for the heat stress equations, the base temperatures had a mean of 5.3 °C, similar to a previously published value of 6 °C. The last variety had noticeably different rates than the others. This study demonstrated that a simple approach (the heat stress equation with one set of parameters for different varieties) should be adequate in many cases. It also demonstrated that more detailed equations can be useful when a more mechanistic system is desired.

**Author Contributions:** Conceptualization, Validation, Investigation, J.R.K.; Methodology and Analysis, J.R.K. and S.K., Data Resources, H.E.Z.T.; Writing-Original Draft Preparation, J.R.K.; Writing-Review and Editing, J.R.K. and S.K.; Visualization, J.R.K. and S.K.

**Funding:** Funding for this research was provided by the USDA ARS Grassland, Soil and Water Res. Lab. USDA is an equal opportunities provider and employer. This work was also supported in part by an appointment to the Agricultural Research Service administered by the Oak Ridge Institute for Science and Education through interagency agreement between the U.S. Department of Energy (DOE) and the U.S. Department of Agriculture (USDA), Agricultural Research Service Agreement #60-3098-5-002.

**Conflicts of Interest:** The authors declare no conflict of interest.

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
