# Peer review of "Back to the Future: Revisiting the Application of an Enzyme Kinetic Equation to Maize Development Nearly Four Decades Later"

_agronomy, doi:10.3390/agronomy9090566_

Round 1

Reviewer 1 Report

Abstract

l. 20. Units

Introduction

l. 103. units

l. 104. DHL is very low.

Table 2. When calculating AIC for PK, were all 6 parameters fitted to the results? If not, then K is too large.

Reviewer 2 Report

I thanks the authors for taking the time to modify the paper.

I found it quite hard to read properly due the track changes visible in the pdf but it seemed to me that the authors had made a detailed and serious efforts to address my comments and suggestions and I am happy to recommend it goes forward for publication.

As a small point I noted in the reply to my specific comments that the authors did not consider that the use of a formal selection criteria framework (i.e. AIC) was appropriate. Nevertheless the paper now seems to make full and appropriate use of exactly this approach, which I believe is an important improvement.

Possibly the formal presentation of AIC and RMSE starting around line 545 is a little unecessary, these are well known statistics and simply referencing relevant sources would be sufficient and more concise. But it is matter of taste.

Author Response

This manuscript is a resubmission of an earlier submission. The following is a list of the peer review reports and author responses from that submission.

Round 1

Reviewer 1 Report

The writing in this paper will have to be greatly improved before consideration for publication. In many cases the text is vague and unclear, and thus requires further elaboration before the results of this paper will be fully understood by readers of the journal. More detailed comments in this regard follow:

Abstract

l. 16: What does ‘not adequate’ mean - predicting anthesis dates or phyllochron intervals? Clarify. Provide results from statistical tests to support this point.

l. 18: How much better? Provide results from statistical tests to support this point. Clearly some improvement can be expected from further fitting.

General: What are the implications of these results for modelling maize phenology?

Introduction

l. 40: Clarify that the Briere model is an alternative to the PK model described previously. Has this model been used in earlier studies?

l. 56: Move this paragraph to p. 1 where the PK model is introduced to improve continuity.

l. 81: Is the ‘broken stick’ an alternative to the 2 models described earlier? Give an equation as was done for these other 2 models.

l. 97 – 102: Move to methods. What are the larger objectives of this study - to improve the robustness with which maize phenology is modelled? What is the importance of such robustness? How does this work build upon earlier modelling studies of maize phenology?

Materials and Methods

l.106. What data? Describe what data was acquired and how these data were measured, and then cite the source. Otherwise the reader has no idea what results will be presented later.

l. 112: What ‘lateral scale’ – time? temperature?

l. 114: This sentence is really unclear – how does a scale shift a size? rephrase

l. 124: What was regressed on what?

l. 128: Avoid repetition. Describe the regressions more clearly.

Results

l. 139: ‘Development rates’ need to be defined. Were these phyllochron numbers relative to maximum value? How were these measured – in a phytotron? What are the temperatures used to estimate these rates – mean daily values? Were these temperatures held constant or did they vary during the day

Fig. 1: How was the water status of these plants maintained – were they frequently irrigated? What if atmospherically induced water stress slowed development at higher temperatures, causing deviations from expected values?

l. 154: cite year

l. 174 – 175: Good point. More points used in fitting reduce error dF in the regressions.

Fig. 4: Indicate results are for the Heat Stress equation without the high temperature data point.

Finally: what are the implications of these findings for modelling maize phenology? How robust are they, and how widely can they be used? What recommendations for modelling maize phenology arise from this paper?

Author Response

The authors have extensively reorganized and reformatting this paper to address the concerns  of this reviewer.  The responses to detailed comments below refer back to the manuscript with changes shown by track changes:

L 16 comment:  This was addressed with new wording on lines 19-20

L 18 comment:  Now addressed on Lines 21-22

General about implications:  Now addressed on lines 31-33

Line 40 comment:  Wording changed on line 76 as suggested

Line 56 comment:  Paragraph now moved to lines 50 to 74 as suggested

LIne 81 comment:  Equations now given on lines 143-155

Line 97-102 comment:  These concerns now addressed 175-179

Line 106 comment:  This now described on lines 183-195

Line 112 comment:  This now addressed on line 198

Line 114 comment:  This now addressed on lines 198-201

Line 124 comment:  This now addressed on lines 219-220

Line 128 comment:  Unclear what this suggestion says.

Line 139 comment:  This now described on lines 230-231

Fig. 1 comment:  This now addressed on line 195

Line 154 comment:  This now correct on line 253

Comment on lines 174-175: Not sure how to respond to this comment

Fig. 4 comment:  This changed on lines 287-288

Final comment about implications:  This now addressed on lines 309-311

Reviewer 2 Report

General

This seems like a useful and interesting topic. There is quite good data set and a potentially useful application of a different models to evaluate which is pragmatically the most applicable. The models include the classic linear response to temperature up to an optimum (heat-stress model as referred to by the authors). This is the bedrock of many crop models so always worth thinking about.

There are lots of different models, some formally presented, some described (?). They are presented as a bit of a chronology of their development. I think it will help to present more in terms of the later results to create a logical structure that applies throughout the paper. To the new reader it comes out as a bit of a jumble.

It might be my slow understanding but I found the paper quite hard to follow in places, and the model inter-comparison is rather too informal for me. I would want to see some clearer statistical comparisons so we can assess the confidence we may have in the different models. The model parameters need to be present to facilitate subsequent application.

I would recommend that the paper deserves consideration after a rewrite and inclusion of a stronger statistical framework for the model inter-comparison.

Specific

12: seem to use the word ‘ssytem’ for what many would call ‘model’?

13: ‘takes one more step’ seems superfluous? Could just say ‘use’.

17: model ‘did much better’. May be true, but I think this needs much more precision and objectivity.

19-25: this text is rather loose, and is hard to follow. It reads as a narrative of tried X, tried Y, went back to X. I suggest a clearer presentation working back from the justified findings to report the relevant parts of what was done.

78-80 getting a bit ‘notey’

81-83 I wonder if the English has gone wrong here?

83 I wonder if the heat stress equation should also be formally expressed. The rest of this paragraph is quite hard to follow (for me at least).

114-119 why is the base temperature not adjustable?

118 I was expecting to have some clear statements of the adjustable model parameters for each model? It is hard to figure out from the description given, in particular I was interested in the number of adjustables. A later table with their values etc would clarify.

135 ‘was not adequate’. For me this statement needs some summary stats.

Maybe the figures can be condensed or summarised? There are quite lot, in some ways it is good, nothing is hidden behind the best examples, but also does take up a lot of room.

170. I read this paper as a model selection exercise (which is good). However any such analysis needs to account for the different number of adjustable parameters in the candidate models (I presume they are different?). Therefore I would expect so see some use of model selection criteria, nothing fancy, AIC for example would be more than sufficient.

I do think some tabulated parameter values for the models are needed, these are relevant for potential applications and SEs values provide confidence (or not) in the use of the models. Some comments on parameter covariance might be helpful.

I may have misunderstood but I expected to see some results fitting across all varieties together, then separately, allowing a formal test of whether the varieties require a significantly different model fit. Maybe that is what Table 1 is doing but if so I don’t see it?

Author Response

The authors have made several changes to improve the logical structure, as suggested.  The responses to all the specific comments below refer to lines in the uploaded manuscript with changes shown.

Comment  on line 12:  "system" changed to "model" on line 13 as suggested

comment on line 13:  wording changed as suggested on line 14

Comment on line 17:  Wording added on lines 21-22

Comment on lines 19-25:  Wording changed on lines 17-22 to address this comment

Comment on lines 78-80:  Not sure what this means

Comment on lines 81-83:  Unclear what this means

Comment on line 83:  Wording added to address this on lines 143-155

Comment on lines 114-119:  This is now addressed on line 201

Comment on line 118:  adjusted parameter values now given with each model's description, on lines 98-109 and lines 64-70

Comments on maybe condensing figures:  I left the figures as in the original.  I feel it is important to show all the details that are in these figures.

Comment on line 170:  While I agree that AIC is useful when comparing complex simulation models, I feel it is beyond the scope of this study

Comment on adding SE values to the parameters:  I believe such added discussion on parameter covariance is beyond the scope of this study.

Comment on results fitting across all varieties together and separately:  We did fitting of each variety separately for the Heat Stress eqauations and it was done for the Briere equations.  However, only one set of parameters was used for the Poikilotherm equation for all varieties..

Round 2

Reviewer 1 Report

 I have reservations about the utility of this work as noted in point 7 below. There are a few points that need to be addressed by the authors.

1.       All values in the paper must have units. E.g. development rates and their RMSE in 1/day, all terms in the poikilothermic eqn,

2.       Is the Briere_2 equation correct? The square root term does not appear to have been replaced.

3.       L. 164, 208: I don’t get why the temperature factor had to be introduced. There is a lot of curve fitting here.

4.       L. 188: avoid repetition.

5.       L. 241: ‘described’

6.       Make sure all Tonnang et al. references are fully cited.

7.       Lastly in the Discussion, the authors need to address some of the causes for inaccuracy of the models apparent in the curve fitting. Crop development rates are governed by their diurnal canopy temperatures, which can differ from diurnal air temperatures which in turn differ a lot from daily mean temperature used in the equations. This is particularly important given the strong nonlinearity of these equations. e.g. are diurnal temperatures varying from 15 to 35 C the same as ones that remain at 25 C in their effects on development? The models would use the same daily means for both. Perhaps inaccuracy in the time scale of the temperature term in these equations is causing some of the problems in curve fitting, such as the use of temperature factors as noted in point 3 above. The authors should point out that there is a need for phenology modelling to move beyond these old approaches from 40 years ago.

8.       The response to my earlier comments refer to lines that no longer exist in the revised manuscript (e.g. l. 309-311).

Author Response

Comment 1.  Now corrected on lines 61-69

Comment 2. The equation was correct but the wording describing it was not correct.  The wording is corrected now on lines 85-86

Comment 3. These issues are now addressed in the rewording in lines 162-166

Comment 4:  This section deleted to avoid repetition (lines 192-196)

Comment 5:  "describe" now "described" on line 245

Comment 6.  Tonnang et al. references now fully cited on lines 246 and 247

Comment 7.  Discussion added to address these issues on lines 284-293
